# Evaluation of Choline Metabolic Genes in the Liver of the Dam as Candidates for Mediating Choline’s Efficacy in Mitigating Ethanol-Induced Cell Death in the Neural Tube: A Preliminary Analysis

**DOI:** 10.3390/genes17010042

**Published:** 2025-12-31

**Authors:** Tasfia Chowdhury, David Ashbrook, Jennifer D. Thomas, Daniel Goldowitz, Kristin Hamre

**Affiliations:** 1Department of Anatomy and Neurobiology, University of Tennessee Health Science Center, Memphis, TN 38163, USA; tchowdh2@uthsc.edu (T.C.); dashbroo@uthsc.edu (D.A.); 2Center for Behavioral Teratology, San Diego State University, San Diego, CA 92182, USA; thomas3@sdsu.edu; 3Department of Medical Genetics, University of British Columbia, Vancouver, BC V6T 1Z4, Canada; dan.goldowitz@ubc.ca

**Keywords:** FASD, alcohol, choline, human, mouse models, polymorphisms

## Abstract

**Background/Objectives**: Emerging evidence has suggested that choline is an effective treatment for at least some of the neurobehavioral deficits associated with Fetal Alcohol Spectrum Disorders (FASD). However, the mechanism of how choline works to ameliorate ethanol’s teratogenic effects, and whether it acts directly on the fetus or indirectly by altering the uterine environment, remains unknown. Previous work from our lab demonstrated that 4 BXD mouse strains that show high levels of ethanol-induced cell death on embryonic day 9.5 (E9.5) have differential responses to choline supplementation. This differential response in mouse strains highlights a need to further understand the role of genetics in choline metabolism. Because the liver is the central organ for choline metabolism, and the embryonic liver of mice is not functional this early in gestation, we focused on choline metabolism in the liver of the dam. **Methods**: Using a bioinformatics approach, the goals were to assess whether (1) genetic differences in liver choline metabolism in the dam could affect ethanol-induced cell death in a genotype-specific manner and (2) any of these candidate genes in the liver of the dam could be linked to differential response to choline amongst the strains. By performing a literature review, haplotype analysis among the 4 BXD strains, and liver protein expression analysis among 3 strains, we show that there are genetic differences in choline metabolic genes that are consistent with the hypothesis that maternal choline metabolism could mediate differential sensitivity. **Results**: While we identified two genes as promising candidates for the variable responses to choline supplementation among the four previously identified BXD strains choline/ethanolamine phosphotransferase 1 (*cept1*) and choline transporter gene solute carrier family 44 member 1 (*slc44a1*), the wealth of data on *slc44a1* makes it the stronger candidate and suggests that it should be further explored. **Conclusions**: Genetic differences in maternal choline metabolism are present and may underlie variable therapeutic responses to choline, warranting a hypothesis that requires further investigation across animal models and human populations.

## 1. Introduction

Fetal alcohol spectrum disorders, or FASDs, define the wide range of symptoms that can be present in a child with prenatal exposure to alcohol [1]. Although certain characteristic physical traits may be seen, patients diagnosed with FASD often display cognitive deficits due to damage to the central nervous system (CNS) during development, leading to substantial effects on learning, behavior, and interpersonal relationships [1,2,3,4]. The true incidence of FASD is unclear, but recent analyses in the United States have identified the prevalence of FASD as being 24 to 48 per 1000 live births [5]. It is currently considered one of the leading causes of preventable cognitive deficits in the developed world [3,6]. The nature of the cognitive deficits is diverse and often differs from individual to individual, but, as shown in both animal and human studies, it can include attention-deficit hyperactivity disorder, learning disabilities, developmental delays, poor coordination and abnormal reflexes, impaired problem-solving, poor social skills, and decreased executive function [2,3,7,8]. These alterations have a substantial impact on quality of life and have been shown to impact an individual’s ability to perform well in school, maintain long-term employment, and increase the chances of participating in high-risk behaviors [3]. Currently, the CDC guidelines for FASD treatment largely revolve around symptomatic management [9]. Pharmacotherapeutics, however, could provide a means to ameliorate some of ethanol’s neuroteratogenic effects.

One potential therapeutic is choline. Choline is an essential nutrient found in meat, eggs, poultry, fish, and dairy that plays a critical role in numerous processes in the body, including DNA methylation, acetylcholine production, and the synthesis of neurotransmitters and phospholipids [10,11]. When choline is absent from the diet, humans show liver damage, skeletal muscle damage, and increased reactive oxygen species, making external consumption essential [12,13,14]. After consumption, choline is transported to the liver, the primary site of choline metabolism, where it can then proceed through multiple pathways for diverse fates in the body [12,13,14]. In addition to metabolism of choline by the liver, during pregnancy, choline is also metabolized by the placenta [15,16,17]. Placental metabolism of choline has been shown to be critically important during late gestation [15,16,17], but its role in early gestation has not been thoroughly investigated. The presence of placental choline transporters at early developmental stages is currently unexplored [15,16,17,18], although it is likely to be highly relevant. Further, additional studies have shown that choline can act indirectly on the embryo by altering various aspects of the uterine environment, such as angiogenesis [19]. Because the role of the placenta in regulating intrauterine choline levels remains unclear, and the liver is not metabolically active in mice until embryonic day 14.5, maternal hepatic metabolism of choline could be a significant modulator of choline levels in the uterine environment during early gestation [20].

Choline is especially important during periods of rapid cell growth, such as pregnancy, because it is essential for building cell membranes. This results in greater utilization of the nutrient during these times. This heightened demand, combined with evidence that alcohol consumption lowers choline levels in the body [21], suggests that alcohol consumption during pregnancy further increases the risk of choline deficiency. Given that choline supplementation improved cognitive function when given in utero and during early childhood, choline deficiency was proposed as a potential contributor to the neurobehavioral deficits in FASD patients [7,8,22,23]. Further, supplementation is being evaluated as a treatment option for pregnant mothers exposed to alcohol and for children diagnosed with FASD [24,25,26]. For example, an initial study found that choline supplementation resulted in improvements in a range of cognitive tests four years after follow-up in children with FASD [26]. In animal models, studies demonstrated improvements in alcohol-related birth and brain weight deficits and alterations in reflex development in animal models, as well as a host of other effects [8,27,28]. A systematic review evaluating 189 abstracts on the impact of choline supplementation in animal models also demonstrated improved molecular and/or epigenetic outcomes in numerous brain regions in animals exposed to alcohol [29]. Physicians are already empirically treating their FASD patients with choline, but specific guidelines on dosage have not been elucidated [30,31]. Significantly, research exploring the efficacy of choline treatment on individuals with FASD has found that certain dosages of choline were not an effective intervention for school-aged children with FASD [25]. In individuals not exposed to ethanol, research has shown that populations with specific genetic polymorphisms in the choline metabolic pathway show an increased response to the supplement compared to individuals without these polymorphisms [32], suggesting that the dose of choline is an important variable in both treatment efficacy and avoidance of untoward adverse effects.

To gain further understanding of the relationship between genetics and choline supplementation efficacy, we previously performed a study comparing the level of cell death in the forebrain and brainstem, two brain areas susceptible to early prenatal ethanol exposure, in 4 strains of alcohol-exposed (B6, BXD51, BXD73, BXD2) mice after choline supplementation [33]. These strains were chosen because of their high sensitivity to the neuroteratogenic effects of alcohol administration early in the formation of the neural tube, as measured by the level of cell death [34]. When choline was administered concurrently with the ethanol, the study found that choline ameliorated ethanol-induced cell death across all 4 strains without causing enhanced cell death in the control mice, who were given choline without ethanol exposure, further demonstrating the efficacy of choline as an FASD treatment option and discounting it as being potentially teratogenic [33]. Further analysis, however, demonstrated that the level of protection afforded by choline varied amongst the strains [33]. This provides evidence that the effective dose of choline is not equivalent across all genotypes. Given the promising role of choline supplementation as a treatment option for children exposed to alcohol in-utero, and the understanding that genetic polymorphisms play a role in its efficacy, the purposes of this study are to (1) assess whether there is evidence of genetic differences in choline metabolic genes in the liver of the dam consistent with the hypothesis that liver metabolism in the dam could be in mediating choline’s efficacy, and (2) determine whether any of these genes are candidates for impacting the efficacy of choline supplementation in these 4 strains of BXD mice and thus, begin to identify a preliminary and hypothesis-generating list of candidate genes. To further assess the genetic contribution to the differential response, the present analyses used bioinformatic analyses to identify potential candidate genes that could mediate this differential response.

## 2. Materials and Methods

In the present study, a sequential analysis was conducted to identify potential candidate genes that alter choline processing in the maternal liver of the mice (See Figure 1). After each analysis, a set of criteria, defined with each bioinformatic step, was used to eliminate genes that were considered less likely to be critical in mediating differential responses. The analysis focused on the parental strains and the BXD recombinant inbred mouse strains derived from parental B6 or D2 strains. Each BXD mouse has a differential contribution of genes from the parental strains and phenotypic differences in particular traits can be used to assess genetic contributions to specific phenotypes [35]. The bioinformatic analysis was performed using Genenetwork, a repository of datasets used to study complex networks of genes, molecules, and higher-order gene function and phenotypes (https://genenetwork.org/).

### 2.1. Previous Study

The cell death values from previously published data were used (33). Briefly, four mouse strains were examined: C57BL/6J (B6), BXD51, BXD73, and BXD2. On embryonic day 9, dams were given either ethanol or isocaloric maltose-dextrin as a caloric control (5.8 g/kg in two administrations separated by 2 h) simultaneously with choline at one of 3 doses: 0 (as a choline-exposure control), 100, or 250 mg/kg. Embryos were collected 7 h after the initial ethanol administrations. Cell death was analyzed using TUNEL staining in the developing forebrain and brainstem. Multiple embryos were analyzed from each litter, and a litter mean was generated. The litter was used as the unit of analysis. From each strain, 5–8 litters were analyzed in each treatment group (ethanol or control) at each choline dose. Strain means, rather than individual litter means, were used to provide more reliable values. Because there was variability between strains in the level of cell death, both after ethanol exposure and in controls (there are low levels of cell death in all control embryos at this stage of development), another value was determined for each strain: mean percent reduction in ethanol-induced cell death. Mean percent cell reduction rather than individual litter data was used because averaging across the litters minimizes variance, corrects for individual litter differences, and provides a more representative estimate of choline’s effect on the forebrain and brainstem across the BXD strains. The mean percent reduction in ethanol-induced cell death was calculated using the cell death levels in the different groups and using the following formula: [(choline + ethanol group)-(maltose/dextrin control groups)} divided by {(ethanol no choline group)-(maltose/dextrin control groups)}. The maltose/dextrin control groups (with and without choline) were combined because they were not significantly different.

### 2.2. Identification of Choline Metabolic Genes Through Literature Review

A literature review was performed to identify genes in the liver that play a significant role in choline metabolism. While choline metabolism can occur in a variety of organ systems, the liver was chosen because it is the primary site of both choline and alcohol metabolism. The literature review was performed using PubMed, Embase, and Google Scholar using the keywords “choline” “metabolism”, and “liver”. Inclusion criteria for the search included papers published in the last 10 years and papers outlining genes implicated in choline metabolism in the liver. Exclusion criteria included papers identifying genes in the choline metabolic pathway outside of the liver and papers published over 10 years prior. Choline metabolism pathways related to other organ systems were excluded.

### 2.3. Haplotype Analysis

The sequence of the previously identified genes was examined in each of the four BXD strains to identify if they expressed the B6 or D2 haplotype. BXD mice have been previously genotyped, and files for the genes are available on the GeneNetwork website. More information on how the coding-region haplotypes were defined and assigned is available at Peirce et al. and Ashbrook et al. [36,37]. Files for the genes identified during the literature review were analyzed to determine if the haplotypic expression of each of the identified genes was B6 or D2. If genes were heterogeneous, they were labeled H, and if they were recombinant, they were labeled R. The following criteria were used: genes that had no haplotypic variation across the four strains, meaning that they all expressed the same haplotype for the gene being explored, were eliminated from further analytic steps, as they were unlikely to be the cause of choline metabolic variability across the four strains.

### 2.4. Liver Protein Analysis

With the liver being the primary source of choline metabolism [11], increased or decreased protein levels produced by the significant genes were hypothesized to significantly increase or decrease the rate of choline metabolism. We prioritized liver proteome data over transcript-level or eQTL measurements because protein is the functional molecule and because protein and mRNA levels are not always consistent. Liver protein levels of the remaining genes were identified using the EPFL/ETHZ BXD Liver Proteome Control Diet (CD) (Nov19) dataset in GeneNetwork (GN) for this analysis. The BXD liver proteome data can be obtained by using the filename EPFL/ETHZ BXD Liver Proteome CD (Nov19) or the accession code: GN888. The data is accessible using the following query in GN:

Species: Mouse

Group: BXD Family

Type: Liver Proteome

Dataset: EPFL/ETHZ BXD Liver Proteome CD (Nov19)

Get Any: Gene name inserted here

Links to the datasets can be found in the Appendix A, and a detailed description of this data can be found in Williams et al. [38]. The liver proteome values were profiled based on the liver proteome of 2157 mice from 89 strains, and the data shown are mean values for the strains. Protein levels over 7 are considered above background as previously described [38], meaning that protein is present within the tissue, and only proteins with liver protein levels above 7 were considered. Variability between the strains was analyzed with the Coefficient of Variation analysis. Nonzero values were considered to represent variation. Protein expression values were given in a logarithmic scale, meaning that differences of 1.0 are equivalent to two times the expression level. The following criterion was used: genes with no liver protein expression variability across the three strains (B6, BXD51, BXD73) were eliminated from further consideration, as equal levels of liver protein expression are likely to correlate with differential metabolic activity.

### 2.5. Correlation Analysis

As a secondary analysis to explore gene variation further, correlation analyses were performed comparing protein expression levels in the liver of the remaining genes with the percent cell death in the E9.5 forebrain and brainstem of each of the three BXD strains with liver proteome data available. The purpose was to determine if differences in protein expression in the liver displayed any trends with regard to differences in cell death identified in the previous study. Four correlation analyses were performed for each gene based on choline dosage and location of cell death: one for 100 mg/kg in the brainstem, one for 250 mg/kg in the brainstem, one for 100 mg/kg in the forebrain, and one for 250 mg/kg in the forebrain. Each BXD data point indicated the mean percent reduction in ethanol-induced cell death among 5–8 litters. Percent reduction in ethanol-induced cell death was obtained and calculated as described above. The Pearson correlation coefficient (R) was found for each analysis, and values over 0.7, a widely accepted cutoff value across biological sciences, were considered to have a strong linear correlation.

### 2.6. Correlation Analysis of SNPs with Phenotypes in Human Databases

As an additional validation step, an analysis was conducted to determine if previous studies have shown a link between SNPs in any of these genes and relevant phenotypes. Genes were analyzed to identify if SNPs in the genes were associated with an increased incidence of alcohol use disorders, developmental disorders, or liver disorders using Genome-Wide Association Studies (GWAS). GWAS catalogs were identified using the Google search engine. Catalogs that contained the genes of interest were included. The GWAS catalogs used for this analysis were the NHGRI-EBI Catalog (https://www.ebi.ac.uk/gwas/ (accessed 1 July 2022), GeneAtlas from the University of Edinburgh (http://geneatlas.roslin.ed.ac.uk/), the GWAS Atlas from the Department of Complex Trait Genetics at VU University Amsterdam (https://atlas.ctglab.nl/), and BioBank Japan PheWeb (https://pheweb.jp/). The NHGRI-EBI Catalog was used as the primary source, as all associations were previously deemed significant with SNP-trait *p*-values < 1.0 × 10^−5^ in the overall population (initial GWAS + replication). The remaining catalogs above were used to explore if further associations were found that were below the NHGRI-EBI Catalog threshold. Within each catalog, each gene was individually searched in the query, and SNPs were screened for alcohol-related disorders, liver-related disorders, and developmental disorders. *p*-values over 0.05 were excluded. In addition to GWAS exploration, a literature review was also conducted to identify if previous studies found SNPs of the genes responsible for different levels of choline metabolism. This literature review included data found prior to October 2023.

## 3. Results

### 3.1. Identification of Choline Metabolic Genes and Haplotype Analysis

Based on the search criteria outlined in the Methods, 22 genes were identified as being relevant to choline metabolism in the liver. These genes were Aldehyde dehydrogenase 7 family member A1 (*Aldh7a1*), Aldehyde dehydrogenase 9 family member A1 *Aldh9a1,* Betaine-Homocysteine Methyltransferase (*Bhmt*), Betaine-Homocysteine Methyltransferase 2 (*Bhmt2*), Choline acetyltransferase (*Chat*), Choline dehydrogenase (*Chdh*), Choline kinase alpha (*Chka*), Choline kinase beta (*Chkb*), choline/ethanolamine phosphotransferase 1 (*Cept1*), choline phosphotransferase 1 (*Chpt1*), phosphate cytidylyltransferase 1A (*Pcyt1a*), phosphate cytidylyltransferase 1B (*Pcyt1b*), 5-methyltetrahydrofolate-homocysteine methyltransferase (*Mtr*), phosphatidylethanolamine N-methyltransferase (*Pemt*), solute carrier family 22 member 1 (*Slc22a1*), solute carrier family 22 member 3 (*Slc22a3*), solute carrier family 5 member 7 (*Slc5a7*), solute carrier family 44 member 1 (*Slc44a1*), solute carrier family 44 member 2 (*Slc44a2*), solute carrier family 44 member 3 (*Slc44a3*), solute carrier family 44 member 4 (*Slc44a4*), and solute carrier family 44 member 5 (*Slc44a5*). Next, the haplotype analysis was conducted to assess whether the sequence of each gene is identical across the strains or if there was sequence variance in a particular gene. The analyses found that 12 genes had haplotypic variability (*Aldh7a1, Bhmt, Bhmt2, Chka, Chkb, Cept1, Chpt1, Slc22a3, Slc5a7, Slc44a1, Slc44a3, and Slc44a5)*. Haplotypic variability was defined as at least 1 haplotypic variation (B6/D2/H/R) among the 4 strains in the coding regions of the genes (See Table 1). Ten genes were removed for having no haplotypic variability *(Aldh9a1, Chat, Chdh, Pcyt1a, Pcyt1b, Mtr, Pemt, Slc22a1, Slc44a2*, and *Slc44a4).*

### 3.2. Liver Protein Analysis Results

Protein levels in the liver were examined to identify genes that were expressed above background levels in the liver and to explore if there were protein level differences in the genes between the BXD strains (See Table 2). The dataset did not include BXD2 data, so this mouse strain was excluded from the analysis from this point forward. Additionally, several of the genes were not present in the database and therefore could not be examined. Only five genes had liver proteome data available and had values above 7 (*Aldh7a1, Bhmt, Bhmt2, Cept1,* and *Slc44a1).* Of the five genes that had significant expression values, only three genes (*Aldh7a1*, *Cept1*, and *Slc44a1*) had coefficient of variation values that were not 0. Two genes (*Bhmt* and *Bhmt2)* were removed from the gene list for having no variation (See Table 3).

### 3.3. Correlation Analysis Results

Correlation analyses were conducted to evaluate the relationship between liver protein expression values and percent reduction in ethanol-induced cell death (See Figure 2, Figure 3 and Figure 4). This step of the analysis was to identify general trends between liver protein expression values and percent reduction in ethanol-induced cell death and was not performed to elucidate the strengths of correlations due to the low sample size. Two genes (*Cept1* and *Slc44a1)* were the only genes that demonstrated a trend between level of protein expression in the liver and percent cell death in the forebrain and brainstem, with at least ¾ of the correlations having Pearson coefficients above 0.7 (See Table 4). For *Cept1*, the brainstem *p*-values ranged from 0.11 to 0.27, and for *Slc44a1*, the brainstem *p*-values ranged from 0.06 to 0.33. The forebrain *p*-values for both genes were not significant. *Aldh7a1* was removed for not demonstrating any positive or negative trends. Positive correlations within the data indicated that increased liver protein expression of the gene could possibly correlate with increased levels of cell death in the brain. Of note, the BXD51 and B6 strains had similar liver protein expression levels of *Cept1*, resulting in their differing percent cell reduction in ethanol-induced cell death data points being plotted for the same liver protein expression value. Further information about raw numeric values used in correlation calculations, including proteome values, percent cell death means per strain, and sample sizes, can be found in the Appendix A.

### 3.4. Functional Analyses of the Candidate Genes

Lastly, literature analyses were performed to assess whether previous studies had detected a relationship between SNPs in any of these genes in either ethanol-related phenotypes or in neurodevelopmental disorders in human populations. Exploration of Genome-Wide Association Studies for the final candidate genes revealed that all four of them were associated with alcohol use disorders, alcoholic liver disease, mental and behavioral disorders due to the use of alcohol, liver disorders causing metabolic dysfunction, mood disorders, or developmental disorders. *Cept1*’s SNP analysis yielded 45 SNPs associated with alcohol use disorders: 5 SNPs associated with liver disorders and 3 SNPs associated with developmental disorders (See Table 5). Additional relevant SNPs were associated with altered gamma glutamyltransferase activity, psychogenic disorders, and random mental health disorders. *Slc44a1*’s SNP analysis yielded 62 SNPs associated with alcohol use disorders: 8 SNPs associated with liver disorders and 8 SNPs associated with developmental disorders (See Table 6). Additional relevant SNPs were related to disorders of lipid metabolism, liver abscesses associated with chronic liver disease, depression, lack of normal physiologic development, autonomic nervous system dysfunction, height, and increased labor and delivery complications. These SNPs within the candidate genes have associations with alcohol processing, neurodevelopment, and behavioral disorders, supporting the hypothesis that these genes are implicated in variable outcomes after developmental alcohol exposure and choline treatment.

## 4. Discussion

The present study evaluated whether there were genetic differences in genes associated with choline metabolism, and, therefore, whether these differences are consistent with the hypothesis that differences in choline metabolism in the liver of the mother could indeed impact the development of the fetus, thereby generating additional hypotheses for subsequent study. Further, two candidate genes were identified as meeting the initial bioinformatic criteria as well as showing strong relationships to other ethanol-related phenotypes and neurodevelopmental disorders. Given that one of the genes, *Slc44a1*, has been shown to be important in choline’s effectiveness in the child [25], it is interesting to further hypothesize that the maternal genotype may also modulate its efficacy in the embryo or fetus, and shows the translational relevance of these studies.

Choline is an essential nutrient that has various functions in the body, including DNA methylation, acetylcholine production, and the synthesis of phospholipids [39]. It has also been shown to regulate aspects of brain development and attenuate neurodevelopmental damage [40]. Given choline’s significance in brain development, and studies that demonstrate lower levels of choline in the brains of healthy women exposed to alcohol, deficiency of the nutrient has been studied as a potential cause of the neurobehavioral deficits seen in FASD patients and supplementation is being implemented as a treatment option for pregnant mothers exposed to alcohol and children with FASD [4,24,25,26,30,41]. The previous study from our lab examined the efficacy of choline in ameliorating ethanol-induced damage on embryonic day 9 in the mouse [33], a time just after neural tube closure. Given that this time period is one of the earliest in terms of brain development, it is likely that at this stage of development, the most important effects of choline would be on either DNA methylation or on lipid biosynthesis. Given that ethanol exposure has previously been shown to affect both of these processes [42,43,44], either or both of these processes could be the mechanisms by which choline provides protection for the developing brain due to ethanol exposure.

However, choline could also have indirect effects on fetal development by altering the uterine environment, i.e., altering aspects outside the embryo itself. Given that the fetal liver is not functioning at this point in development [20,45], alterations to the uterine environment provide a strong potential mechanism. Previous research in other systems has provided two potential mechanisms. First, choline has been shown to alter the development of the placenta. For example, in a mouse model of gestational diabetes, as well as in human placental cells in vitro, choline supplementation specifically altered various morphologic aspects of the placenta and its cells [46]. Second, choline supplementation during pregnancy has been shown to alter angiogenesis in both the embryo and the placenta [19]. Studies have also shown significant changes in gene expression in angiogenic pathways in placentas following choline supplementation [16,19]. These suggest that evaluation of choline’s indirect effects on the fetus is an important aspect to be explored, particularly this early in development.

With choline proving itself to be a promising therapeutic for FASD, and the rise in prenatal supplements containing choline, dosing guidelines must be elucidated to increase efficacy and minimize risk [47]. Excessive choline is associated with an increased risk of heart disease, GI discomfort, body odor, low blood pressure, and liver damage [48]. Previous research has demonstrated that different individuals process choline differently, meaning that there are differences in choline requirements within the general population [49,50]. Identifying which genes contribute to these different requirements will lead to more targeted therapy [25,50]. Certain polymorphisms related to choline have already been associated with variable responses to choline, like SNPs of *Ckha* and *Slc44a1.* This study aimed to further identify candidate genes implicated in the metabolism of choline and that could be integral to providing more targeted choline dosing.

From the analysis conducted in the present study, the final candidate genes were *Cept1* and *Slc44a1.* The SNP analyses combined with the literature review showed that polymorphisms of both genes were associated with alcohol use disorders, liver disorders, and developmental disorders within human populations. Choline metabolism underlies the effects of alcohol on the body, leading to the hypothesis that SNPs associated with increased adverse effects due to alcohol consumption could potentially cause an increased risk of adverse effects in the mother and ultimately the fetus during gestation [51]. Furthermore, SNPs demonstrating the impact of altered choline processing on developmental traits, both physical and mental, indicate that varied choline metabolism is a potential mechanism for abnormal fetal development in FASD. Choline also plays a significant role in liver protein expression through its role as a methyl donor [52]. These proteins are crucial for physiologic activities and development [52]. Altered levels of protein expression from the liver highlight how different levels of choline processing could impact the typical course of these functions. The association between polymorphisms of the candidate genes with developmental disorders was particularly interesting because abnormalities in fetal and early childhood development are at the core of FASDs [1]. As shown in this study, associated developmental disorders include mental disorders, behavioral disorders, and disorders related to physiologic development, possibly drawing a link between FASD and the functionality of these genes in the perinatal period.

*Cept1*, or choline/ethanolamine phosphotransferase 1, assists in the synthesis of phospholipids that contain choline or ethanolamine [53]. It catalyzes the last step of the biosynthesis of phosphatidylcholine and phosphatidylethanolamine by transferring the substituted phosphate group from CDP-choline/ethanolamine to diacylglycerol [54]. Previous enzyme assays and metabolic labeling with radiolabeled choline demonstrated that loss of *Cept1* dramatically decreases choline phospholipid biosynthesis and that its function is essential for the activation of transcription factors in the liver, like PPAR-alpha, making variations in this gene a promising cause of differing phenotypes attributable to alterations in choline metabolism [53]. While entirely preliminary in nature, the results suggest the possibility that higher liver expression of this molecule could increase cell death rather than decrease it. However, given the low sample size, this interpretation should be taken with extreme caution.

*Slc44a1* encodes solute carriers that assist in the transmembrane transport of choline [55]. Slc44a1 is located in multiple parts of the cell, including the cytosol, mitochondria, and nucleoplasm [55]. The SNPs of *Slc44a1* were associated with differing dietary choline requirements among different races [50]. Furthermore, a study exploring FASD symptom improvement with choline supplementation found that 14 to 16 Slc44a1 SNPs were associated with greater symptom improvement compared to the control, indicating that the polymorphisms in this gene play a significant role in the human body’s response to choline supplementation [25]. Interestingly, this gene shows a positive correlation with higher levels of the protein correlated with greater reductions in cell death. Of note, while higher maternal liver protein expression levels of *Slc44a1* may correlate with more negative therapeutic effects of choline, this interpretation should be taken with caution because of the low sample size. Further, other interpretations and potential confounders are possible. For example, some of the effects may also be due to differences in maternal metabolic states or placental transport capacity.

In the present study, we focused on the choline metabolism that occurs in the mother. At this developmental stage in mice, the liver is rudimentary and minimally active, if at all, as shown in previous work in animals and humans [45]. Therefore, the fetal liver is unlikely to significantly contribute to choline metabolism at this developmental stage. Previous research in children with FASD has focused on the genotype of the child rather than the mother [25,26,50], but the present study emphasizes the importance of the maternal genotype as well. Additionally, while we are focused on how these genes behave independently of one another, it is very possible that these candidate genes may interact with one another and act synergistically. Future work should include further testing to uncover which SNPs of the candidate genes are most significantly associated with varied choline supplementation responses in FASD patients and then testing different choline dosages in individuals with different SNPs. In animal models, studies could include examining mice with maternal conditional knockout of the genes in question: *Cept1* or *Slc44a1*. Future studies should also include ways to assess how these genes interact with each other.

A limitation of this study is that the only organ explored was the liver. Choline is also metabolized in the kidney and placenta, so candidate genes involving the kidney could have also played a significant role in the supplementation’s differential response [12]. Furthermore, only 3 BXD mice were explored, and liver proteome data were only available for five genes of interest in the liver protein analysis, so associations were based on 3 data points and a limited number of genes in the choline metabolic pathway, making the present study underpowered, preliminary, and exploratory in nature. In the future, this analysis should be repeated with more BXD strains to increase the power of the data and should include additional genes in the choline metabolic pathway. In addition, BXD mice maternal and offspring genomes are perfectly correlated, while human maternal and offspring genomes are not [35]. Previous research on the impact of these candidate genes on human choline processing helps mitigate this concern. In the future, further research using reciprocal crosses or embryo transfers would be beneficial to help distinguish maternal hepatic effects from fetal genotype effects. Furthermore, based on previous analyses, there is evidence to suggest that choline is associated with various brain, alcohol, liver, and developmental phenotypes, as were evaluated in the present study. Population stratification and heterogeneity, however, may influence the GWAS associations referenced in this study. Because many choline-related GWAS findings are primarily derived from adult cohorts, the extent to which these associations truly reflect prenatal choline metabolism, fetal liver function, or embryonic developmental pathways is uncertain. Thus, the genetic relationships discussed here should be interpreted as hypothesis-generating and must be further established by future studies. Finally, reliance on public proteome datasets introduces inherent technical limitations, because the liver protein expression profiles in the reference mice may differ subtly from those in the mice examined in our study.

Overall, identifying the genes involved in the choline metabolic pathway that significantly alter choline requirements will play an important role in the future safety and efficacy profile of choline as a therapeutic for FASD. FASD has a major impact on the trajectory of a person’s life and leads to lifelong struggles with intellectual pursuits, occupations, and interpersonal relationships. This, coupled with the fact that a critical period of brain development in a fetus occurs prior to 6 weeks of gestation, a time when many women are unaware that they are pregnant, makes finding a therapeutic paramount [56]. Choline offers a promising solution to this growing and almost unavoidable problem, and further elucidating its safety profile and dosing is integral to its role in combating this public health concern. This study provides a preliminary and hypothesis-generating analysis of potential candidates that could be important in mediating its efficacy when given to pregnant mothers.

## Figures and Tables

**Figure 1 genes-17-00042-f001:**
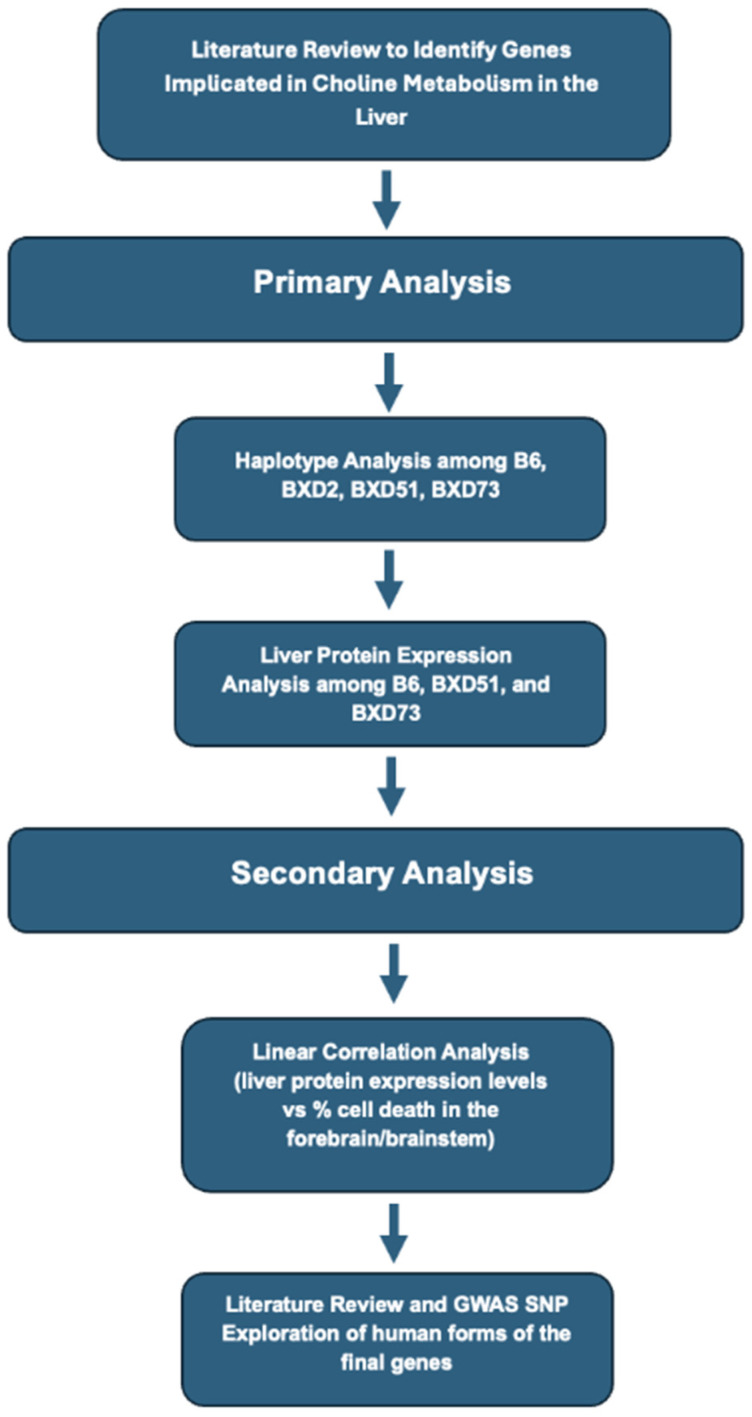
Flowchart of sequential analysis conducted to identify potential candidate genes.

**Figure 2 genes-17-00042-f002:**
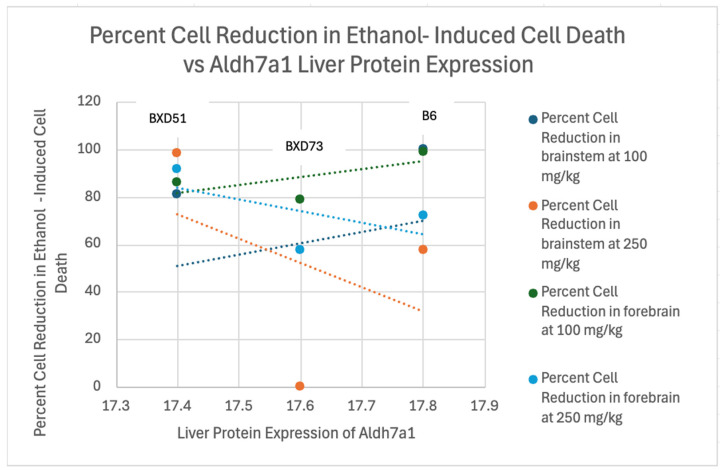
Linear Correlation Analysis of Percent Reduction in Ethanol-Induced Cell Death in the Brainstem and Forebrain of B6, BXD51, and BXD73 mice vs. Liver Protein Expression of Aldh7a1. There was little relationship between Aldh7a1 protein expression and cell death suggesting that maternal levels of this protein were unlikely to be critical.

**Figure 3 genes-17-00042-f003:**
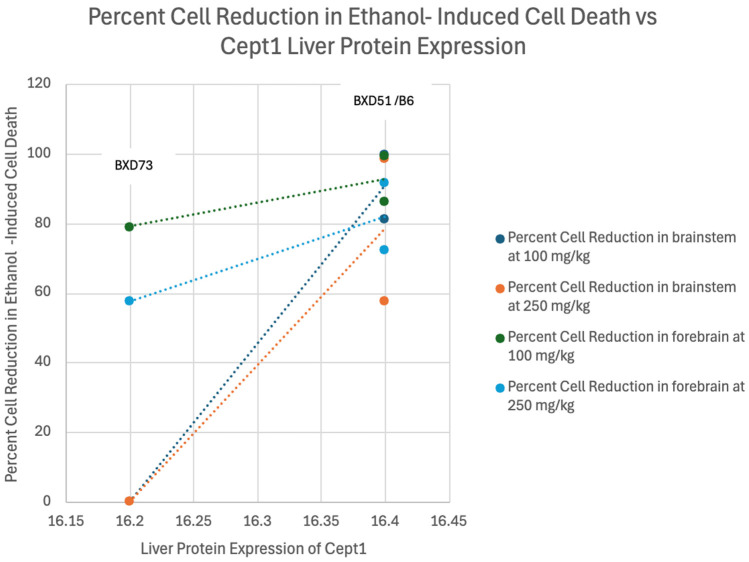
Linear Correlation Analysis of Percent Reduction in Ethanol-Induced Cell Death in the Brainstem and Forebrain of B6, BXD51, and BXD73 mice vs. Liver Protein Expression of *Cept1*. There is a significant correlation, although the small number of data points suggests caution.

**Figure 4 genes-17-00042-f004:**
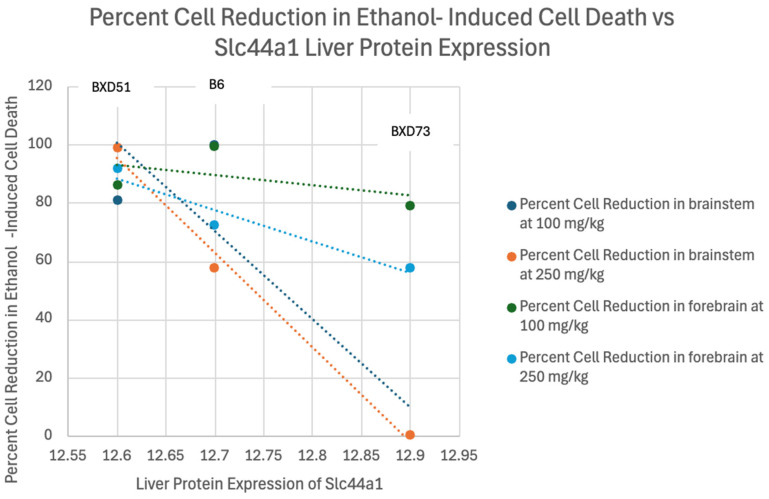
Linear Correlation Analysis of Percent Reduction in Ethanol-Induced Cell Death in the Brainstem and Forebrain of B6, BXD51, and BXD73 mice vs. Liver Protein Expression of *Slc44a1*.

**Table 1 genes-17-00042-t001:** Haplotype Analysis of Identified Genes of Interest (B6: Haplotype is that of B6 mice, D2: Haplotype is that D2 mice, NV: No variation in haplotype amongst the 4 strains, R: Haplotype had recombinant expression, H: Haplotype had heterozygous expression); Genes that showed no haplotype variation are not bolded while genes that showed haplotype variation are bolded.

	B6	BXD51	BXD73	BXD2
** *Aldh7a1* **	**B6**	**B6**	**B6**	**D2**
*Aldh9a1*	B6	B6	B6	B6
** *Bhmt* **	**B6**	**B6**	**B6**	**D2**
** *Bhmt2* **	**B6**	**B6**	**B6**	**D2**
*Chat*	NV	NV	NV	NV
*Chdh*	NV	NV	NV	NV
** *Chka* **	**B6**	**B6**	**R**	**B6**
** *Chkb* **	**H**	**H**	**B6**	**B6**
** *Cept1* **	**B6**	**D2**	**D2**	**D2**
** *Chpt1* **	**B6**	**D2**	**B6**	**D2**
*Pcyt1a*	B6	B6	B6	B6
*Pcyt1b*	NV	NV	NV	NV
*Mtr*	NV	NV	NV	NV
*Pemt*	NV	NV	NV	NV
*Slc22a1*	NV	NV	NV	NV
** *Slc22a3* **	**B6**	**D2**	**D2**	**B6**
** *Slc5a7* **	**B6**	**B6**	**D2**	**D2**
** *Slc44a1* **	**B6**	**D2**	**D2**	**B6**
*Slc44a2*	B6	B6	B6	B6
** *Slc44a3* **	**B6**	**D2**	**B6**	**D2**
*Slc44a4*	NV	NV	NV	NV
** *Slc44a5* **	**B6**	**R**	**R**	**R**

**Table 2 genes-17-00042-t002:** Liver protein levels of identified genes of interest (numerical values represent liver protein expression levels of each gene in the liver proteome of the different strains of BXD mice, n/a was entered when the data was unavailable); Eliminated genes are not bolded. Bolded genes are potential candidates.

	B6	BXD51	BXD73	BXD2
** *Aldh7a1* **	**17.8**	**17.4**	**17.6**	**n/a**
*Bhmt*	22	22	22	n/a
*Bhmt2*	17	17	17	n/a
** *Cept1* **	**16.4**	**16.4**	**16.2**	**n/a**
** *Slc44a1* **	**12.7**	**12.6**	**12.9**	**n/a**

**Table 3 genes-17-00042-t003:** Coefficient of Variation Analysis of each gene in the liver proteome of the different strains of BXD mice, n/a was entered when the data was unavailable; eliminated genes are not bolded. Bolded genes are potential candidates.

	Coefficient of Variation
**Aldh7a1**	**1.136**
Bhmt	0
Bhmt2	0
**Cept1**	**0.707**
**Slc44a1**	**1.200**

**Table 4 genes-17-00042-t004:** Pearson Coefficient (R) values of the identified genes of interest based on location and choline dosage. Bolded genes are potentially candidates while those that are not bolded are not perspective candidate.

	Brainstem 100 mg/kg	Brainstem 250 mg/kg	Forebrain 100 mg/kg	Forebrain 250 mg/kg
*Aldh7a1*	.085	.268	.636	.571
** *Cept1* **	**.996**	**.963**	**.771**	**.821**
** *Slc44a1* **	**.913**	**.998**	**.521**	**.962**

**Table 5 genes-17-00042-t005:** Gene variant, affected phenotype, and *p*-value of *Cept1* SNPs.

Gene Variant	Phenotype	*p*-Value
rs10494127	K70 Alcoholic liver disease	3.01 × 10^−2^
rs57171998	K70 Alcoholic liver disease	3.32 × 10^−2^
rs75484460	K70 Alcoholic liver disease	3.37 × 10^−2^
rs78334840	K70 Alcoholic liver disease	3.47 × 10^−2^
rs4839101	K70 Alcoholic liver disease	3.53 × 10^−2^
rs1591456	K70 Alcoholic liver disease	3.62 × 10^−2^
rs7514887	K70 Alcoholic liver disease	3.91 × 10^−2^
rs2278578	K70 Alcoholic liver disease	3.95 × 10^−2^
rs56273356	K70 Alcoholic liver disease	4.13 × 10^−2^
rs67493833	K70 Alcoholic liver disease	4.23 × 10^−2^
rs6662965	K70 Alcoholic liver disease	4.24 × 10^−2^
rs61804623	K70 Alcoholic liver disease	4.33 × 10^−2^
rs61804560	K70 Alcoholic liver disease	4.34 × 10^−2^
rs56296601	K70 Alcoholic liver disease	4.34 × 10^−2^
rs4839098	K70 Alcoholic liver disease	4.41 × 10^−2^
rs11805357	K70 Alcoholic liver disease	4.46 × 10^−2^
rs12406191	K70 Alcoholic liver disease	4.53 × 10^−2^
rs10494126	K70 Alcoholic liver disease	4.63 × 10^−2^
rs116484160	K70 Alcoholic liver disease	4.69 × 10^−2^
rs56740658	K70 Alcoholic liver disease	4.69 × 10^−2^
rs7521727	K70 Alcoholic liver disease	4.72 × 10^−2^
rs61808584	K70 Alcoholic liver disease	4.77 × 10^−2^
rs55857873	F10 Mental and behavioral disorders due to use of alcohol	6.78 × 10^−3^
rs626339	F10 Mental and behavioral disorders due to use of alcohol	1.29 × 10^−2^
rs2243393	F10 Mental and behavioral disorders due to use of alcohol	1.44 × 10^−2^
rs183103888	F10 Mental and behavioral disorders due to use of alcohol	1.45 × 10^−2^
rs9429589	F10 Mental and behavioral disorders due to use of alcohol	1.48 × 10^−2^
rs2580040	F10 Mental and behavioral disorders due to use of alcohol	1.49 × 10^−2^
rs325922	F10 Mental and behavioral disorders due to use of alcohol	1.49 × 10^−2^
rs325923	F10 Mental and behavioral disorders due to use of alcohol	1.53 × 10^−2^
rs192792696	F10 Mental and behavioral disorders due to use of alcohol	2.20 × 10^−2^
rs325929	F10 Mental and behavioral disorders due to use of alcohol	2.34 × 10^−2^
rs182732	F10 Mental and behavioral disorders due to use of alcohol	2.53 × 10^−2^
rs532249550	F10 Mental and behavioral disorders due to use of alcohol	2.62 × 10^−2^
rs35041930	F10 Mental and behavioral disorders due to use of alcohol	2.66 × 10^−2^
rs573629054	F10 Mental and behavioral disorders due to use of alcohol	3.25 × 10^−2^
rs146158422	F10 Mental and behavioral disorders due to use of alcohol	3.52 × 10^−2^
Affx-4271322	F10 Mental and behavioral disorders due to use of alcohol	3.59 × 10^−2^
rs608881	F10 Mental and behavioral disorders due to use of alcohol	3.77 × 10^−2^
rs78334840	F10 Mental and behavioral disorders due to use of alcohol	3.80 × 10^−2^
rs7521727	F10 Mental and behavioral disorders due to use of alcohol	3.92 × 10^−2^
rs657605	F10 Mental and behavioral disorders due to use of alcohol	4.17 × 10^−2^
rs629619	F10 Mental and behavioral disorders due to use of alcohol	4.35 × 10^−2^
rs675874	F10 Mental and behavioral disorders due to use of alcohol	4.94 × 10^−2^
rs11590090	Alcoholism	4.28 × 10^−2^
rs1335645	gamma glutamyltransferase activity	1.82 × 10^−19^
rs2296380	gamma glutamyltransferase activity	9.00 × 10^−10^
rs4839104	gamma glutamyltransferase activity	2.00 × 10^−20^
rs77355087	gamma glutamyltransferase activity	3.00 × 10^−21^
rs1335645	gamma glutamyltransferase activity	4.00 × 10^−18^
rs2885805	Psychogenic Disorders	8.02 × 10^−3^
rs958798	schizophrenia	4.85 × 10^−3^
rs17036350	random mental disorders	4.67 × 10^−2^

**Table 6 genes-17-00042-t006:** Gene variant, affected phenotype, and *p*-value of *Slc44a1* SNPs.

Gene Variant	Phenotype	*p*-Value
rs12686004	alcohol-related disorders	1.54 × 10^−2^
rs2575876	alcohol-related disorders	3.02 × 10^−2^
rs149604610	K70 Alcoholic liver disease	6.94 × 10^−3^
rs112519618	K70 Alcoholic liver disease	1.45 × 10^−2^
rs146633331	K70 Alcoholic liver disease	1.64 × 10^−2^
rs185740504	K70 Alcoholic liver disease	1.65 × 10^−2^
rs116909302	K70 Alcoholic liver disease	1.69 × 10^−2^
rs80013298	K70 Alcoholic liver disease	1.70 × 10^−2^
rs187474970	K70 Alcoholic liver disease	1.70 × 10^−2^
rs74409690	K70 Alcoholic liver disease	1.78 × 10^−2^
rs181319399	K70 Alcoholic liver disease	1.79 × 10^−2^
rs188419287	K70 Alcoholic liver disease	1.79 × 10^−2^
rs138278086	K70 Alcoholic liver disease	1.81 × 10^−2^
rs137912394	K70 Alcoholic liver disease	1.82 × 10^−2^
rs78364414	K70 Alcoholic liver disease	1.82 × 10^−2^
rs143012032	K70 Alcoholic liver disease	1.82 × 10^−2^
rs116866483	K70 Alcoholic liver disease	1.83 × 10^−2^
rs142915188	K70 Alcoholic liver disease	1.84 × 10^−2^
rs117347201	K70 Alcoholic liver disease	1.84 × 10^−2^
rs143713468	K70 Alcoholic liver disease	1.85 × 10^−2^
rs117205874	K70 Alcoholic liver disease	1.86 × 10^−2^
rs191075086	K70 Alcoholic liver disease	1.86 × 10^−2^
rs151236148	K70 Alcoholic liver disease	1.87 × 10^−2^
rs117398101	K70 Alcoholic liver disease	1.87 × 10^−2^
rs117111328	K70 Alcoholic liver disease	1.87 × 10^−2^
rs118046379	K70 Alcoholic liver disease	1.88 × 10^−2^
rs12335477	K70 Alcoholic liver disease	2.02 × 10^−2^
rs191661087	K70 Alcoholic liver disease	2.31 × 10^−2^
rs568438809	K70 Alcoholic liver disease	2.35 × 10^−2^
rs117589569	K70 Alcoholic liver disease	3.29 × 10^−2^
rs71494515	K70 Alcoholic liver disease	3.65 × 10^−2^
rs36019711	K70 Alcoholic liver disease	3.68 × 10^−2^
rs35413260	K70 Alcoholic liver disease	3.70 × 10^−2^
rs7028183	K70 Alcoholic liver disease	3.73 × 10^−2^
rs35825389	K70 Alcoholic liver disease	3.74 × 10^−2^
rs1397666	K70 Alcoholic liver disease	3.77 × 10^−2^
rs13285648	K70 Alcoholic liver disease	3.78 × 10^−2^
rs74807089	K70 Alcoholic liver disease	4.04 × 10^−2^
rs7022737	K70 Alcoholic liver disease	4.33 × 10^−2^
rs7021567	K70 Alcoholic liver disease	4.33 × 10^−2^
rs10120572	K70 Alcoholic liver disease	4.33 × 10^−2^
rs10118989	K70 Alcoholic liver disease	4.34 × 10^−2^
rs71494519	K70 Alcoholic liver disease	4.38 × 10^−2^
rs185433179	K70 Alcoholic liver disease	4.38 × 10^−2^
rs1397665	K70 Alcoholic liver disease	4.41 × 10^−2^
rs36112131	K70 Alcoholic liver disease	4.41 × 10^−2^
rs71494516	K70 Alcoholic liver disease	4.41 × 10^−2^
rs71494518	K70 Alcoholic liver disease	4.41 × 10^−2^
rs13299679	K70 Alcoholic liver disease	4.43 × 10^−2^
rs113771472	K70 Alcoholic liver disease	4.44 × 10^−2^
rs71494517	K70 Alcoholic liver disease	4.45 × 10^−2^
rs13286370	K70 Alcoholic liver disease	4.47 × 10^−2^
rs10114041	K70 Alcoholic liver disease	4.59 × 10^−2^
rs35412407	K70 Alcoholic liver disease	4.89 × 10^−2^
rs145172265	F10 Mental and behavioral disorders due to use of alcohol	1.04 × 10^−3^
rs139882511	F10 Mental and behavioral disorders due to use of alcohol	1.29 × 10^−2^
rs74849027	F10 Mental and behavioral disorders due to use of alcohol	3.39 × 10^−2^
rs28503319	F10 Mental and behavioral disorders due to use of alcohol	3.42 × 10^−2^
rs62575094	F10 Mental and behavioral disorders due to use of alcohol	3.85 × 10^−2^
rs10481710	F10 Mental and behavioral disorders due to use of alcohol	4.33 × 10^−2^
rs72744260	F10 Mental and behavioral disorders due to use of alcohol	4.55 × 10^−2^
rs117265373	F10 Mental and behavioral disorders due to use of alcohol	4.81 × 10^−2^
rs4742971	Disorders of lipid metabolism	4.31 × 10^−2^
rs2515629	secondary liver malignant neoplasm of the liver	4.91 × 10^−3^
rs4743034	Liver abscess and sequelae of chronic liver disease	2.49 × 10^−2^
rs7032034	Total cholesterol levels	1.00 × 10^−22^
rs374445	Triglyceride levels	4.00 × 10^−9^
rs11506820	Low density lipoprotein cholesterol levels	5.00 × 10^−8^
rs7859070	High density lipoprotein cholesterol levels	2.00 × 10^−18^
rs16924584	Total cholesterol levels	1.00 × 10^−20^
rs4742971	Depression	4.31 × 10^−2^
rs7861820	Lack of normal physiological development	5.38 × 10^−3^
rs4149268	Pathological, developmental or recurrent dislocation	2.11 × 10^−2^
rs649891	lack of normal physiology	3.26 × 10^−2^
rs117827556	labor and delivery complications	2.36 × 10^−5^
rs2515629	Disorders of the autonomic nervous system	4.74 × 10^−2^
rs7861820	Dementia with cerebral degenerations	4.58 × 10^−2^
rs7031663	height	4.00 × 10^−10^

## Data Availability

All data is freely available in GeneNetwork.org.

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
