# Peer review of "Evaluation of Choline Metabolic Genes in the Liver of the Dam as Candidates for Mediating Choline’s Efficacy in Mitigating Ethanol-Induced Cell Death in the Neural Tube: A Preliminary Analysis"

_genes, 2025, doi:10.3390/genes17010042_

Round 1

Reviewer 1 Report

Comments and Suggestions for Authors

Choline has been studied as a supplement for ameliorating cognitive/behavioral deficits in fetal alcohol spectrum disorders caused by prenatal ethanol exposure.  The efficacy of such an approach is mixed.  Genetic differences have been considered the underlying cause of differential choline treatment effects. Using different strains of mice, the authors reveal a possible link between genetic differences in maternal liver enzymes involved in choline metabolism and ethanol-exposure-induced cell death.  The manuscript is well written, with an adequate literature review of choline metabolism and the selection of genes.  The MS can be improved by the following.

  1. Details of the ethanol and choline treatment regimen should be provided in the methodology session. The collection of endpoints should also be described.
  2. The primary issue is the use of the average protein level and the reduction of brain cells in each strain. This approach reduces the number of data points and statistical power.  In addition, individual differences in maternal liver function and brain cell reduction within each litter within each strain could provide valuable information on the association between brain cell reduction and liver protein expression.  The correlation for each strain should also be provided.
  3. There are no controls for choline exposure (the zero choline group). It is not clear how the percent cell reduction is calculated.
  4. Only three mouse strains are used. The Pearson correlation coefficient is effective only for detecting a linear relationship and is unstable with very few data points.   Other statistical analyses should also be considered.
  5. The figures are hard to understand, especially Figure 5. Correlational coefficients based on 2 data points are not very meaningful.

Author Response

Comment 1: “Details of the ethanol and choline treatment regimen should be provided in the methodology session. The collection of endpoints should also be described.”

Response 1: Thank you for pointing this out. We agree with this comment and have added in a section titled Previous Study on Page 5, Paragraph 1, Line 145.

Previous Study:

The cell death from previously published data was used (33). Briefly, four mouse strains were examined:  C57BL/6J (B6), BXD51, BXD73, BXD2. On embryonic day 9, dams were given either ethanol or isocaloric maltose-dextrin as a caloric control (5.8 g/kg in two administrations separated by 2 h) simultaneously with choline at one of 3 doses: 0 (as a choline-exposure control), 100 or 250 mg/kg. Embryos were collected 7 h after the initial ethanol administrations. Cell death was analyzed using TUNEL staining in the developing forebrain and brainstem. Multiple embryos were analyzed from each litter, and a litter mean was generated and the litter used as the unit of analysis. From each strain, 5-8 litters were analyzed at in each treatment group (ethanol or control) at each choline dose. Strain means, rather than individual litter means, were used to provide more reliable values. Because there was variability between strains in the level of cell death, both after ethanol exposure and in controls (there is low levels of cell death in all control embryos at this stage of development) another value was determined for each strain: mean percent reduction in ethanol-induced cell death. Mean percent cell reduction rather than individual litter data was used because averaging across the litters minimizes variance, corrects for individual litter differences, and provides a more representative estimate of choline’s effect on the forebrain and brainstem across the BXD strains. The mean percent reduction in ethanol-induced cell death was calculated using the cell death levels in the different groups and using the following formula: [(choline + ethanol group)-(maltose/dextrin control groups)} divided by {(ethanol no choline group)-(maltose/ dextrin control groups)}. The maltose/ dextrin control groups (with and without choline) were combined because they were not significantly different.  

Comment 2: "The primary issue is the use of the average protein level and the reduction of brain cells in each strain. This approach reduces the number of data points and statistical power.  In addition, individual differences in maternal liver function and brain cell reduction within each litter within each strain could provide valuable information on the association between brain cell reduction and liver protein expression.  The correlation for each strain should also be provided."

Response 2: Thank you for pointing this out. We agree with this comment. As per reviewer 2, we have changed the section on the correlation to state that this analysis is preliminary and hypothesis-generating. We have, additionally, changed the other portions of the paper to reflect this. We also added the following statement on Page 5, Paragraph 1, Line 160.

“Mean percent cell reduction rather than individual litter data was used because averaging across the litters minimizes variance, corrects for individual litter differences, and provides a more representative estimate of choline’s effect on the forebrain and brainstem across the BXD strains.”

Comment 3: "There are no controls for choline exposure (the zero choline group). It is not clear how the percent cell reduction is calculated."

Response 3: Thank you for pointing this out. We agree with this comment and have added in: “The mean percent reduction in ethanol-induced cell death was calculated using the cell death levels in the different groups and using the following formula: [(choline + ethanol group)-(maltose/dextrin control groups)} divided by {(ethanol no choline group)-(maltose/ dextrin control groups)}. The maltose/ dextrin control groups (with and without choline) were combined because they were not significantly different.” on Page 5, Paragraph 1, Line 163. We have also clarified (see Preliminary Studies) that both an ethanol without choline group was used as control for the ethanol exposure and a maltose/dextrin plus choline was used as as a control for choline alone. 

Comment 4: "Only three mouse strains are used. The Pearson correlation coefficient is effective only for detecting a linear relationship and is unstable with very few data points.   Other statistical analyses should also be considered."

Response 4: Thank you for pointing this out. We agree with this comment and have clarified that the statistical analysis was only meant to show a general trend among the data points rather than identify the strength of correlation on Page 6, Paragraph 3, Line 224: “The purpose was to determine if differences in protein expression in the liver displayed any trends with regard to degree of correlation with differences in cell death identified in the previous study.” and “Correlation analyses were conducted to evaluate the relationship between liver protein expression values and cell death levels. This step of the analysis was to identify general trends between liver protein expression values and cell death levels and was not performed to elucidate the strengths of correlations. Two genes (Cept1 and Slc44a1) were the only genes that demonstrated a trend correlation between level of protein expression in the liver and percent cell death in the forebrain and brainstem, with at least ¾ of the correlations having Pearson coefficients above 0.7. Aldh7a1 was removed for not demonstrating any positive or negative trends. having this linear correlation. Positive correlations within the data indicated that increased liver protein expression of the gene could possibly correlate with increased levels of cell death in the brain brain cell death.” on Page 9, Paragraph 1, Line 308

Comment 5: "The figures are hard to understand, especially Figure 5. Correlational coefficients based on 2 data points are not very meaningful."

Response 5: Thank you for pointing this out. We agree with this comment and have done the following to address this concern:

  • Clarified that one of the analyses had 2 strains with the same liver protein expression in the results section. 
  • Clarified the analyses used to generate the figures and clarified the legends on the axes (See Previous Study section above).
  • Added more information in the figure legends.
  • On Page 9, Paragraph 1, Line 308: “Correlation analyses were conducted to evaluate the relationship between liver protein expression values and cell death levels. This step of the analysis was to identify general trends between liver protein expression values and cell death levels and was not performed to elucidate the strengths of correlations. Two genes (Cept1 and Slc44a1) were the only genes that demonstrated a trend correlation between level of protein expression in the liver and percent cell death in the forebrain and brainstem, with at least ¾ of the correlations having Pearson coefficients above 0.7. Aldh7a1 was removed for not demonstrating any positive or negative trends. having this linear correlation. Positive correlations within the data indicated that increased liver protein expression of the gene could possibly correlate with increased levels of cell death in the brain brain cell death.” “Of note, the BXD51 and B6 strains had similar liver protein expression levels of cept1, resulting in their differing percent cell reduction in ethanol-induced cell death data points being plotted for the same liver protein expression value.”
  • Fixed figure 3 to be more representative of the data.

Reviewer 2 Report

Comments and Suggestions for Authors

Please review the attached review report for the authors' revision.

Author Response

Comment 1: "For Method transparency and reproducibility, provide the exact GeneNetwork dataset IDs, versions, and query parameters used for haplotype, proteome, and expression extraction, so readers can reproduce the steps by providing the accession/link and date accessed."

Response 1: Thank you for pointing this out. We agree with this comment. We added the following on Page 6, Paragraph 1, Line 192: “Liver protein levels of the remaining genes were identified using the EPFL/ETHZ BXD Liver Proteome Control Diet (CD) (Nov19) dataset in GeneNetwork (GN) for this analysis. The BXD liver proteome data can be obtained by using the filename EPFL/ETHZ BXD Liver Proteome CD (Nov19) or the accession code: GN888. The data is accessible using the following query in GN:

Species: Mouse

Group: BXD Family

Type: Liver Proteome

Dataset: EPFL/ETHZ BXD Liver Proteome CD (Nov19)

Get Any: *insert gene name*”

  • We also added links to the datasets in the supplementary files.

Comment 2: Clearly specify how coding-region haplotypes were defined and assigned from GeneNetwork.

Response 2: Thank you for pointing this out. We agree with this comment. We added the following on page 5, paragraph 3, line 183: “More information on how the coding-region haplotypes were defined and assigned by using SNPs to sequence across the genome. Further information about how this was performed is available at Peirce et al. and Ashbrook et al. [36, 37].”

Comment 3: "In the method section, report the sample sizes behind the liver proteome values (how many biological replicates per strain) and whether those proteome values are means, medians, or individual measurements."

Response 3: We agree with this comment. We added the following on page 6, paragraph 2, line 207: “Links to the datasets can be found in the supplementary files, and a detailed description of this data can be found in Williams et al. [38]. The liver proteome values were profiled based on the liver proteome of 2157 mice from 89 strains, and the data shown are mean values for the strains.”

Comment 4: "The liver proteome analysis uses only three strains (B6, BXD51, BXD73) because BXD2 data were unavailable. Discuss in detail how conclusions based on correlations from n = 3 strains are inherently underpowered and exploratory. Reframe any causal language to be explicitly preliminary."

Response 4: Thank you for pointing this out. We agree with this comment. We added the following on page 19, paragraph 4, line 515: “Furthermore, only 3 BXD mice were explored, and liver proteome data were only available for five genes of interest in the liver protein analysis, so associations were based on 3 data points and a limited number of genes in the choline metabolic pathway, making the present study underpowered, preliminary, and exploratory in nature. In the future, this analysis should be repeated with more BXD strains to increase the power of the data and should include additional genes in the choline metabolic pathway”

Comment 5: "Where possible, expand the analysis to include additional BXD strains with available proteome or transcriptome data, or provide a rationale why the three strains used are representative. If additional data are impossible, clearly state that the analyses are hypothesis-generating."

Response 5: Thank you for pointing this out. We agree with this comment. We added the following on page 20, paragraph 2, line 547: “This study provides a preliminary and hypothesis-generating analysis of potential candidates that could be important in mediating its efficacy when given to pregnant mothers.”

Comment 6: "The Pearson correlation threshold of R > 0.7 is reasonable as an exploratory filter but must be accompanied by p-values given the tiny sample size (n = 3). For n = 3, virtually any correlation coefficient is unstable; provide exact p-values and discuss the statistical limitations."

Response 6: Thank you for pointing this out. We agree with this comment. We added the following on page 9, paragraph 1, line 309: “Correlation analyses were conducted to evaluate the relationship between liver protein expression values and cell death levels. This step of the analysis was to identify general trends between liver protein expression values and cell death levels and was not performed to elucidate the strengths of correlations, as an n=3 makes virtually any correlation coefficient unstable. Two genes (Cept1 and Slc44a1) were the only genes that demonstrated a trend correlation between level of protein expression in the liver and percent cell death in the forebrain and brainstem, with at least ¾ of the correlations having Pearson coefficients above 0.7. For Cept1, the brainstem p-values ranged from 0.11 to 0.27, and for Slc44a1, the brainstem p-values ranged from 0.06 to 0.33. The forebrain p-values for both genes were not close to significant. Aldh7a1 was removed for not demonstrating any positive or negative trends. having this linear correlation. Positive correlations within the data indicated that increased liver protein expression of the gene could possibly correlate with increased levels of cell death in the brain brain cell death.” Of note, the BXD51 and B6 strains had similar levels of liver protein expression levels of cept1, resulting in only two data points used for the analysis. “

Comment 7: "Reconsider interpretation of positive correlations: the manuscript implies that higher maternal liver protein expression correlates with increased brain cell death (i.e., worse outcomes). Discuss alternative interpretations and potential confounders."

Response 7: Thank you for pointing this out. We agree with this comment. We added the following on page 19, paragraph 2, line 493: “Of note, while higher maternal liver protein expression levels of slc44a1 may correlate with more negative therapeutic effects of choline, this interpretation should be taken with caution because of the low sample size. Further, other interpretations and potential confounders are possible. For example, some of the effects may also be due to differences in maternal metabolic states or placental transport capacity.”

Comment 8: "Explain why proteome measures (log scale values from GeneNetwork) were prioritized over transcript-level data or eQTLs. If transcript data are available for the liver in GeneNetwork consider repeating correlations using RNA expression to bolster the evidence."

Response 8: Thank you for pointing this out. We agree with this comment. We added the following on page 6, paragraph 1, line 194: “We prioritized liver proteome data over transcript-level or eQTL measurements because protein is the functional molecule and because protein and mRNA levels are not always consistent.”

Comment 9: "Provide the exact GWAS sources, search dates, and query strings for the SNP associations reported. For each candidate gene, report how SNPs were selected and whether associations reported are genome-wide significant"

Response 9: Thank you for pointing this out. We agree with this comment. We added the following on page 7, paragraph 1, line 238: “As an additional validation step, an analysis was conducted to determine if previous studies have shown a link between SNPs in any of these genes and relevant phenotypes. Genes were analyzed to identify if SNPs in the genes were associated with an increased incidence of alcohol use disorders, developmental disorders, or liver disorders using Genome-Wide Association Studies (GWAS). GWAS catalogs were identified using the Google search engine. Catalogs that contained the genes of interest were included. The GWAS catalogs used for this analysis were the NHGRI-EBI Catalog (https://www.ebi.ac.uk/gwas/), GeneAtlas from the University of Edinburgh (http://geneatlas.roslin.ed.ac.uk/), the GWAS Atlas from the Department of Complex Trait Genetics at VU University Amsterdam (https://atlas.ctglab.nl/), and BioBank Japan PheWeb (https://pheweb.jp/). The NHGRI-EBI Catalog was used as the primary source as all associations were previously deemed significant with SNP-trait p-values <1.0x10-5  in the overall population (initial GWAS + replication). The remaining catalogs above were used to explore if further associations were found that were below the NHGRI-EBI Catalog threshold. Within each catalog, each gene was individually searched in the query and SNPs were screened for alcohol-related disorders, liver-related disorders, and developmental disorders. P-values over 0.05 were excluded. In addition to GWAS exploration, a literature review was also conducted to identify if previous studies found SNPs of the genes responsible for different levels of choline metabolism. This literature review included data found prior to October 2023.

Comment 10: "Several p-values in the tables appear inconsistent (e.g., rs56740658 p = 4.69E-020 is likely an error; rs142915188 p = 1.84E4-02 has formatting problems). Carefully check all SNP p-values, ensure scientific notation is correct, and report consistent precision."

Response 10: Thank you for pointing this out. We agree with this comment. This has been corrected.

Comment 11: "Add a brief discussion about population stratification, heterogeneity, and whether the GWAS associations (often discovered in adult populations) are relevant to prenatal choline metabolism and embryonic outcomes."

Response 11: Thank you for pointing this out. We agree with this comment. We added the following on page 19, paragraph 4, line 526: Furthermore, based on previous analyses, there is evidence to suggest that choline is associated with various brain, alcohol, liver, and developmental phenotypes, as are being evaluated in the present study. Population stratification and heterogeneity, however, may influence the GWAS associations referenced in this study. Because many choline-related GWAS findings are primarily derived from adult cohorts, the extent to which these associations truly reflect prenatal choline metabolism, fetal liver function, or embryonic developmental pathways is uncertain. Thus, the genetic relationships discussed here should be interpreted as hypothesis-generating and must be further established by future studies. Finally, reliance on public proteome datasets introduces inherent technical limitations, because the liver protein expression profiles in the reference mice may differ subtly from those in the mice examined in our study.”

Comment 12: "The Discussion should more clearly separate maternal hepatic effects from placental and fetal contributions. Because maternal and offspring genomes are correlated in BXD matings, explain how the authors propose to distinguish maternal hepatic effects from fetal genotype effects in future experiments (such as reciprocal crosses, embryo transfer, maternal knockdown)."

Response 12: Thank you for pointing this out. We agree with this comment. We added the following on page 19, paragraph 4, line 524: “In addition, BXD mice maternal and offspring genomes are perfectly correlated, while human maternal and offspring genomes are not [35]. Previous research on the impact of these candidate genes in human choline processing helps mitigate this concern. In the future, further research using reciprocal crosses or embryo transfers would be beneficial to help distinguish maternal hepatic effects from fetal genotype effects.”

Comment 13: "Recommend and discuss functional validation experiments to test the maternal role directly (examples: maternal-specific knockdown/conditional knockout of Cept1 or Slc44a1, pharmacokinetic measurements of maternal and uterine choline after supplementation, placental transport assays)."

Response 13: Thank you for pointing this out. We agree with this comment. We added the following on page 19, paragraph 3, line 510: “Future work should include further testing to uncover which SNPs of the candidate genes are most significantly associated with varied choline supplementation responses in FASD patients and then testing each SNP with different choline dosages. This could include studies that have maternal conditional knockout of the genes in question: cept1 or slc44a1.”

Comment 14: "Expand the limitations section to explicitly acknowledge (a) the exploratory nature of analyses with very small n, (b) exclusion of other choline-metabolizing organs (kidney, placenta), (c) potential strain differences in choline pharmacokinetics or maternal behavior, and (d) reliance on public proteome datasets with their own technical limitations."

Response 14: Thank you for pointing this out. We agree with this comment. We added the following on page 19, paragraph 4, line 513:A limitation of this study is that the only organ explored was the liver. Choline is also metabolized in the kidney and placenta, so candidate genes involving the kidney could have also played a significant role in the supplementation’s differential response [12]. Furthermore, only 3 BXD mice were explored, and liver proteome data were only available for five genes of interest in the liver protein analysis, so associations were based on 3 data points and a limited number of genes in the choline metabolic pathway, making the present study underpowered, preliminary, and exploratory in nature. In the future, this analysis should be repeated with more BXD strains to increase the power of the data and should include additional genes in the choline metabolic pathway. In addition, BXD mice maternal and offspring genomes are perfectly correlated, while human maternal and offspring genomes are not [35]. Previous research on the impact of these candidate genes in human choline processing helps mitigate this concern. In the future, further research using reciprocal crosses or embryo transfers would be beneficial to help distinguish maternal hepatic effects from fetal genotype effects. Finally, reliance on public proteome datasets introduces inherent technical limitations, because the liver protein expression profiles in the reference mice may differ subtly from those in the mice examined in our study.”

Comment 15: "Improve figure quality and readability (axes, units, strain labels, regression lines with confidence bands). For correlation plots, plot the raw data points and annotate n and Pearson R (with p-value)."

Response 15: Thank you for pointing this out. We agree with this comment and have corrected this. P-values were not included as the plots were not meant to portray correlation but rather a general trend.

Comment 16: "In Table 1, bolding and NV/R/H labels are somewhat confusing. Add a short legend clarifying each label and consider adding a column that explicitly states whether haplotypic variation passed the inclusion filter."

Response 16: Thank you for pointing this out. We agree with this comment and have added this in.

Comment 17: "Provide supplemental files with the exact queries, scripts (R or Python) used for analyses, raw numeric values used in correlation calculations (proteome values, percent cell death means per strain, and sample sizes). If these cannot be included in a repository, clearly state how reviewers can access them."

Response 17: Thank you for pointing this out. We agree with this comment. We added the following on page 9, paragraph 1, line 324: “Further information about raw numeric values used in correlation calculations, including proteome values, percent cell death means per strain, and sample sizes can be found in the supplementary files.”

Comment 18: "Justify why protein expression > 7 (GeneNetwork log scale) was considered “above background” and explain whether that threshold is dataset-specific. Provide a reference or documentation from the proteome dataset to support using 7 as a cutoff."

Response 18: Thank you for pointing this out. We agree with this comment. We added the following on page 6, paragraph 2, line 207: “Links to the datasets can be found in the supplementary files, and a detailed description of this data can be found in Williams et al. [38] The liver proteome values were profiled based on the liver proteome of 2157 mice from 89 strains, and the data shown are mean values for the strains. Protein levels over 7 are considered above background as previously described [38], meaning that protein is present within the tissue and only proteins with liver protein levels above 7 were considered.”

Comment 19: "There are minor typographical and formatting errors, such as missing parentheses in "FASD.", inconsistent spacing and punctuation, and reference numbering issues like "9. Bestry M., Symons M .. Larcombe A." that need review."

Response 19: Thank you for pointing this out. We agree with this comment and have corrected this.

Comment 20: "Ensure consistent use of gene nomenclature (mouse gene symbols are italicized in many journals; check MDPI Genes style). Spell out genes at first mention (e.g., CEPT1 (Cept1))."

Response 20: Thank you for pointing this out. We agree with this comment and have corrected this.

Comment 21: "In the Abstract, explicitly state the number of strains analyzed and that the analysis is preliminary/hypothesis-generating"

Response 21: Thank you for pointing this out. We agree with this comment and have adjusted the abstract to read as the following:

Background/Objectives: Emerging evidence has suggested that choline is an effective treatment for at least some of the neurobehavioral deficits associated with Fetal Alcohol Spectrum Disorders (FASD. However, the mechanism of how choline works to ameliorate ethanol’s teratogenic effects, and whether it acts directly on the fetus or indirectly by altering the uterine environment, remains unknown. Previous work from our lab demonstrated that 4 BXD mouse strains that show high levels of ethanol-induced cell death on embryonic day 9.5 (E9.5) have differential responses to choline supplementation. This differential response in mouse strains highlights a need to further understand the role of genetics in choline metabolism. Because the liver is the central organ for choline metabolism, and the embryonic liver of mice is not functional this early in gestation, we focused on choline metabolism in the liver of the dam. Methods: Using a bioinformatics approach, our goals were to assess whether 1) genetic differences in liver choline metabolism in the dam could affect ethanol-induced cell death in a genotype-specific manner and 2) any of these candidate genes in the liver of the dam could be linked to differential response to choline amongst the strains. By performing a literature review and haplotype analysis among the 4 BXD strains, and liver protein expression analysis among 3 strains, we show that there are genetic differences in choline metabolic genes that are consistent with the hypothesis that maternal choline metabolism could mediate differential sensitivity. ResultsWhile we identified two genes as promising candidates for the variable responses to choline supplementation among the four previously identified BXD strains (choline/ethanolamine phosphotransferase 1 (cept1) and choline transporter gene solute carrier family 44 member 1 (slc44a1)), the wealth of data on slc44a1 make it the stronger candidate and suggest that should be further explored. Conclusions: Genetic differences in maternal choline metabolism and may underlie variable therapeutic responses to choline, warranting a hypothesis that requires further investigation across animal models and human populations.”

Round 2

Reviewer 2 Report

Comments and Suggestions for Authors

The authors have corrected as suggested in the peer review report. At this point article is ready to move for further consideration.